# Influence of Exposure Parameters and Implant Position in Peri-Implant Bone Assessment in CBCT Images: An In Vitro Study

**DOI:** 10.3390/jcm11133846

**Published:** 2022-07-02

**Authors:** Paweł Sawicki, Piotr Regulski, Artur Winiarski, Paweł J. Zawadzki

**Affiliations:** 1Department of Cranio-Maxillofacial Surgery, Oral Surgery and Implantology, Faculty of Dental Medicine, Medical University of Warsaw, 02-005 Warsaw, Poland; kontakt@pawelsawicki.pl (P.S.); kcst@wum.edu.pl (P.J.Z.); 2Department of Dental and Maxillofacial Radiology, Faculty of Dental Medicine, Medical University of Warsaw, 61 Zwirki i Wigury Street, 02-091 Warsaw, Poland; 3Department of Dental Propaedeutics and Prophylaxis, Faculty of Dental Medicine, Medical University of Warsaw, 02-006 Warsaw, Poland; artur.winiarski@wum.edu.pl

**Keywords:** CBCT, dental implants, buccal bone, peri-implant artifacts, dentistry

## Abstract

The aim of this study was to assess the impact of dimensional distortion and its changes with modification of exposure setting parameters on the measurements of peri-implant bone margin. Ten titanium dental implants (InKone Primo, Global D, Paris, France) were placed in two prepared bovine ribs. Two bone models and an implant-with-transfer model were scanned with 3shape E4 (3shape, Copenhagen, Denmark) laboratory scanner. Cone beam computed tomography (CBCT) images of two bone models were taken with different values of voltage (60, 70, 80, 90 kV), tube current (4, 10 mA) and voxel size (200, 300 µm). All the data were superimposed using planning software, and the measurements of buccal bone thickness in two selected regions were performed both using CBCT and scan cross-sections. The mean squared error (MSE) being the squared differences between measurements was used in the accuracy assessment of the CBCT device. A one-way ANOVA revealed significant differences between voltage and MSE (*p* = 0.044), as well as implant position and MSE (*p* = 0.005). The distortions of measurements depend on bone margin thickness, and the higher the distance to measure, the higher the error. Accurate measurements of buccal bone thickness (MSE below 0.25) were achieved with voltage values of 70, 80, and 90 kV.

## 1. Introduction

A decrease in the horizontal and vertical dimensions of the alveolar ridge is observed one year after tooth extraction due to bone resorption, which may be exacerbated by inflammatory processes of endodontic or periodontal origin [1,2].

Bone remodelling following tooth extraction can be partially reduced by procedures, such as alveolar ridge preservation, aiming to maintain the bone volume required for implant-supported prosthetic restoration [3,4,5]. When planning such treatment, a minimum of 1.5 mm of buccal bone width surrounding the implant should be provided, which according to Monje et al. is a key factor for favourable long-term aesthetic and functional outcome in dental implant treatment [6].

Due to the observed crestal bone loss subsequent to implant insertion, which averages 0.24mm ± 0.62mm after one year, a follow-up is mandatory to monitor soft and hard peri-implant tissues. Consensus reports recommend periodontal examination and charting (the presence/absence of bleeding and suppuration on probing and probing depth), as well as taking standardized intraoral radiographs [7,8,9]. 

Intraoral (IO) and panoramic radiographs are most commonly used during follow-up due to their low radiation dose and cost-effectiveness. Despite the undeniable advantages of the aforementioned two-dimensional (2D) imaging modalities, they do not allow the assessment of the buccal bone, which is crucial for successful implant treatment. It is possible to evaluate the buccal bone level and its thickness by bone sounding using a periodontal probe, which is an invasive procedure, as well as ultrasonography and cone beam computed tomography (CBCT), which is a three-dimensional (3D), non-invasive imaging modality [10,11,12].

Several studies recommend the use of CBCT for early evaluation of periodontal bone defects. However, due to artifact formation around dental implants, few studies recommend CBCT for bone assessment around dental implants [12,13,14,15,16,17]. Contemporary recommendations for the clinical use of CBCT in implant dentistry developed by Jacobs et al. (2018) indicate that although intraoral radiographs are still considered to be the primary tool for postoperative implant monitoring, we should realize that we need to evaluate three-dimensional bone healing, including morphological, volumetric, and trabecular remodelling [18]. Current guidelines recommend further work on the development of 3D imaging techniques, such as CBCT, which will allow for accurate measurements of peri-implant bone tissue [18,19]. Screening for peri-implant defects at early stages would allow for adequate treatment, such as debridement of the implant surface, bone, and/or soft tissue augmentation procedures or even removal of the dental implant [12,18,20].

Dimensional distortion is a common artifact in dental–maxillofacial radiology. Its presence has been confirmed on both periapical and panoramic radiographs [21,22,23]. Surprisingly, dimensional changes were also observed in CBCT, which is considered one of the most accurate techniques in dental radiology [24]. These distortions may be related to beam hardening, partial volume effects, and metallic artifact reduction algorithms [25]. 

Shape distortions are most common with metal and high radiosensitivity materials and depend on CBCT exposure parameters, such as voltage and field of view. Reduction of both voltage and field-of-view diminishes the amount of beam hardening artifacts and, at the same time, increases the radiation dose. Modification of other exposure parameters, such as tube current or voxel size, affects the amount of noise and radiation dose [26]. Once an appropriate field-of-view has been set up in accordance with the ALARA radiation safety principle (As Low As Reasonably Achievable), it remains necessary to set the remaining exposure parameters [27]. There is a need to find the optimal voltage, tube current, and voxel size settings that allow for the smallest possible measurement error in peri-implant bone assessment while maintaining a low radiation dose.

Reliance on dimensional distortion in the assessment of peri-implant tissues in CBCT images may lead to misdiagnosis and clinically unjustified treatment, which may even result in the deterioration of peri-implant tissues (i.e., peri-implant soft tissue dehiscence and deterioration of aesthetics) [28].

Therefore, the aim of this study was to assess the impact of shape distortion on the measurements of the bone margin surrounding dental implants. The effect of voltage, current, and voxel size was taken into consideration. The null hypothesis was that there was no significant difference between measurement error and voltage, current, implant position, or voxel size.

## 2. Materials and Methods

Two blocks of bovine ribs, obtained from a local slaughterhouse, were prepared and denuded from soft tissues. Bone materials were classified as medium dense (D2–D3) based on the section of fresh bovine rib, drilling resistance during implant bed preparation, and primary implant stability. In D2 type of bone, there is thick dense to porous cortical bone on the crest and coarse trabecular bone within. In D3 type of bone, there is thin, porous cortical bone on the crest and fine trabecular bone within. The D2 bone type is most suitable for implant placement and postoperative healing [29,30]. Ten implant site osteotomies were performed according to the manufacturer’s instructions with a final 3.4 mm drill. Five dental implants (InKone Primo ⌀ = 3.5 mm L = 8.5 mm, Global D, France) were placed into each fresh bovine rib (10 implants in total). Dental titanium implants used in this study are designed with an internal 8-degree conical connection and internal hex. Implants were placed with bone margin thickness: 0.0 mm, 0.2 mm, 0.3 mm, 0.4 mm, 0.5 mm, 0.6 mm, 0.7 mm, 0.9 mm, 1.1 mm, and 1.2 mm, respectively. X-ray markers made of dental composite were attached to each side of the bone model for accurate superimposition. Closed-tray impression transfers were attached to the implants (Figure 1 and Figure 2).

An implant model consisting of the same dental implant with closed-tray impression transfer was also prepared (Figure 2). All metallic materials were coated with scan spray (Renfert, Germany) prior to scanning.

Two bone models and an implant-with-transfer model were scanned with 3shape model E4 (3shape, Denmark). The STL files of the scanned models were exported. Cone beam computed tomography images of two bone models were taken using Vatech Pax-i 3D (Vatech, Hwaseong, Korea). Sixteen CBCTs were taken of each implant with the following exposure parameters of voltage: 60, 70, 80, 90 kV; current: 4, 10 mA; and voxel size: 200, 300 µm. Metal artifact reduction was not used. In total, 160 separate images of dental implants were obtained. DICOM data of CBCT images were exported. Projects containing each bone model (STL), implant-with-transfer model (STL), and bone CBCT images (DICOM) were created using BlueSkyPlan planning software (Blue Sky Bio, Libertyville, IL, USA). All the data were superimposed using planning software and adjusted manually afterwards. A cross-section in the middle of each implant was obtained and CBCT scans were assessed using bone window settings (Figure 3).

Bone margin was measured in BlueSkyPlan planning software on cross sections using a digital distance measure tool in a horizontal plane at two levels: implant neck (L1) and 3.5 mm apically to implant neck (L2). The measurements on the scans were treated as ground truth. The measurements on CBCT images were performed at the same levels as ground truth. The mean squared error (MSE) being the squared differences between measurements was used in the accuracy assessment of CBCT. All measurements were performed twice. The interval between the first and the second reading was at least 6 weeks. The interval between first and second reading was 6 weeks to ensure that the measurements were repeatable and that there was no effect of the first reading on the second one. The intrarater reliability was assessed with the intra-class correlation coefficient (ICC) based on a 2-way mixed-effects mean-rating model. The mean from two measurements was taken into consideration in further statistical analysis. Study design is presented on the flowchart (Figure 4).

A one-way analysis of variance (ANOVA) was used to assess the relationship between voltage and MSE, as well as implant position and MSE. Student’s t-tests were performed for the assessment of current, voxel size, and MSE relationships. Linear regression was used to find the linear equation of relationship between the bone thickness and MSE for each voltage. According to this equation, the range of bone thicknesses was selected for each the MSE was less than 0.25 mm^2^, which corresponds to the error of measurements of less than 0.5 mm. 

## 3. Results

The mean values from ground truth measurements at both bone levels are presented in Table 1. The intrarater reliability was excellent in terms of repeatability of measurements (ICC = 94.1%).

A significant relationship between the voltage and MSE of bone thickness measurement was observed (ANOVA, F = 2.75, *p* = 0.044), proving that the higher the voltage, the lower the MSE. Therefore, a higher voltage is better to reduce the measurement error caused by dental implant material (Table 2).

The relationship between implant position and MSE was also significant (ANOVA, F = 2.74, *p* = 0.005). No statistically significant results were observed for current (*t*-test, *p* = 0.956) or voxel size (*t*-test, *p* = 0.055) (Table 3).

The linear regression analysis revealed that accurate results for bone margin thickness can be obtained for voltage of 70, 80, and 90 kV. The higher the distance to measure, the higher the error. For 60 kV, the MSE is always above 0.25 in the measured range. The higher the voltage, the higher the threshold value of accurate measured distance and the more accurate measurements are feasible (Table 4).

## 4. Discussion

This study confirms the thesis that dimensional error in the measurement of buccal bone thickness around dental implant depends on the voltage and dental implant position. The MSE decreases statistically significantly with increasing voltage and buccal bone width. There were statistically significant differences between current and voxel size values.

These results could have significant clinical implications. During follow-up of patients treated with dental implants, CBCT assessment raises concerns regarding the peri-implant bone thickness and the number of dental artifacts. This study suggests the possibility of decreasing the X-ray dose by setting parameters that do not affect the MSE, such as the voxel size and the intensity of the X-ray tube, at a level that allows the maximal reduction of the X-ray dose [26]. In addition, the awareness of limitations in peri-implant tissue measurement using CBCT requires thorough clinical examination before any intervention is undertaken.

We chose bovine ribs for this in vitro study to simulate alveolar bone. This type of human bone simulation was used previously in the literature on peri-implant bone defects or X-ray artifacts surrounding dental implants [29,31,32,33,34,35]. Bovine rib bone has cortical and cancellous bone of similar thickness and structure as a human mandible [36,37]. In their study, Bredbenner et al. assessed insertion and pull-out torque for 1.0 mm and 2.4 mm outer diameter screws for different substitutes for human cadaveric bone in maxillofacial rigid-fixation research. Although no single material was ideal, it was found that bovine rib could be the material of choice to simulate human cadaveric bone, but statistically significant differences (*p* < 0.05) were found between bovine bone and cadaveric group for pull-out strength [38]. The analysis of artifacts related to the different shape of the human mandible requires further studies. However, the effect of artifacts from the opposite side of the mandible is negligible compared to the beam hardening effect associated with the implant [32].

To the best of our knowledge, there are no studies with comparable measuring methods where the superimposition of the STL models and CBCT images were performed. In the available literature, measurements of buccal bone were performed by bone sounding using a blunt needle or measurement of buccal bone before implant placement or after implant removal [39,40]. The applied methodology allows for measurements of the implant, osseous tissue, and soft tissues in a selected area at any time that has passed since the scans and CBCT were performed. In the methods used until now, these were possible for a limited period of time due to the possibility of performing measurements only prior to implant placement, and the possible damage to the material during storage or after freezing. The main disadvantage of this approach using superimposition is the presence and intensity of artifacts related to X-ray markers or titanium abutments. Image distortion prevents automatic superimposition and requires manual adjustment, which extends the time of project preparation and may decrease its accuracy.

There have been several studies evaluating the accuracy of measurements around dental implants with similar methodologies involving measurements at the same level of bone plate thickness on a model and using CBCT. Wang et al. (2013) used CBCT to perform radiographic images of pigs’ jaws with placed implants. They were cut at every implant site in the bucco-oral direction resulting in 40-μm sections that were stained with toluidine blue, measured, and then compared to CBCT images. Accuracy of −0.22 ± 0.77 mm for measurements on CBCT was observed [41]. Vanderstuyft et al. showed that there is a doubtful zone around a dental implant of about 0.45 mm. This means that buccal bone width below 0.45 mm may not always be observed in CBCT. Implant blooming percentage of up to 12–15% (increase of implant width in CBCT image) and an underestimation of the peri-implant buccal bone thickness, depending on the CBCT device used, by an average of 0.27 ± 0.19 mm (Accuitomo^®^ 170, J. Morita, Kyoto, Japan) and 0.22 ± 0.17 mm (NewTom^®^ VGi evo^®^ (QR Verona, Verona, Italy) were found [39].

González-Martín et al. performed a study with three different computed tomography studies (1 CT and 2 CBCT devices) and reported that for sites with 0.5 mm buccal bone thickness, the probability of being radiographically visible was less than 20% and the odds of bone identification increased for a 1 mm increase in bone thickness. The mean distortion error for all CBCT devices was 0.39 mm [40]. There were no significant differences among the three devices [40]. Rezavi et al. observed an underestimation of buccal bone when it was thinner than 0.8 mm for both selected CBCT devices [42].

Based on the literature, we set the mean squared error at 0.25 in an arbitrary way for the statistical assessment of the dimensional distortion error under selected conditions. The MSE of 0.25 mm^2^ corresponds to the 0.5 mm error in buccal bone measurement between a scan and the CBCT image.

Crestal bone loss following implant placement, as mentioned, is 0.24 ± 0.62 mm after one year. In the context of the results obtained, it is extremely important to properly set the exposure parameters when performing CBCT to determine the thickness of the vestibular bone plate. Despite high voltage value in a CBCT scan, in cases with thin buccal bone plate associated with severe bone loss, the presence of dimensional distortion error may suggest its complete resorption. However, in the case of thick buccal bone plate exposed with low value of voltage, the mean dimensional error may be greater than 0.5 mm, which can significantly distort the measurement result. In any of these situations, a significant measurement error of the buccal bone plate may lead the dentist to make an ill-informed decision about the need for surgical intervention to improve the amount of peri-implant tissue. Vanderstuyft et al. suggest measurements of the crestal peri-implant buccal bone thickness during implant placement surgery. With the baseline buccal bone thickness, implant diameter, and average implant blooming percentage or mean dimensional error as a reference, the subsequent decision on possible intervention is facilitated [39].

This study has several limitations that should be taken into consideration. The present study was conducted with a single CBCT device using several exposure parameters such as voltage, tube current, and voxel size. A study with the use of more CBCT devices might yield different results, especially when considered that the results of the relationship between voxel size and MSE (*p* = 0.055) were close to achieving a level of statistical significance. Secondly, no simulation of soft tissues was performed in this study. The novel methodology used in this study could be refined with the addition of a thin layer of wax to simulate soft tissues. This will require a digital scan of the model with and without a wax layer to further evaluate the ability to measure the thickness of soft and hard tissues. 

The superimposition in BlueSkyPlan planning software could be done automatically or based on observer-defined landmarks (at least five). The best superimposition results could be obtained when landmarks on scan and X-ray models are repeatable and selected in three axes (transversal, antero-posterior, and vertical), keeping an appropriate distance between them. At first, in a pilot study, a scanbody was selected as an abutment attached to each implant instead of a closed-tray impression transfer. The main advantage of a scanbody is its matte surface, which facilitated the digital scanning, but there were issues with superimposition caused by its shape. The shape of a scanbody developed by the manufacturer of the implants used in the present study lacked defined edges enough to determine the same landmarks on a scan and X-ray model due to the artifacts caused by beam hardening on the X-ray model, as the scanbody was also made from titanium. A change from the scanbody to a closed-tray impression transfer and coating it with scanspray partially solved this problem, but it needed manual adjustment due to the presence of artifacts. Superimposition could be improved by selecting different abutment material, which should generate less artifacts in the CBCT image. For instance, it could be made individually using CAD/CAM from radiolucent material, such as polyetheretherketone (PEEK), with radiopaque x-ray markers [43].

Further studies are needed to verify these findings in a larger group of implants using different CBCT machines, exposure parameters (i.e., field of view), and considering the application of fresh human cadaver heads.

## 5. Conclusions

The voltage has an important impact on the accuracy of CBCT measurements. The higher the voltage, the lower the mean squared error of the measurements. The distortions of measurements depend on the thickness of the bone margin, and the higher the distance to measure, the higher the error. Bone margin thickness can be measured accurately (with an MSE of less than 0.25) for 70, 80, and 90 kV.

## Figures and Tables

**Figure 1 jcm-11-03846-f001:**
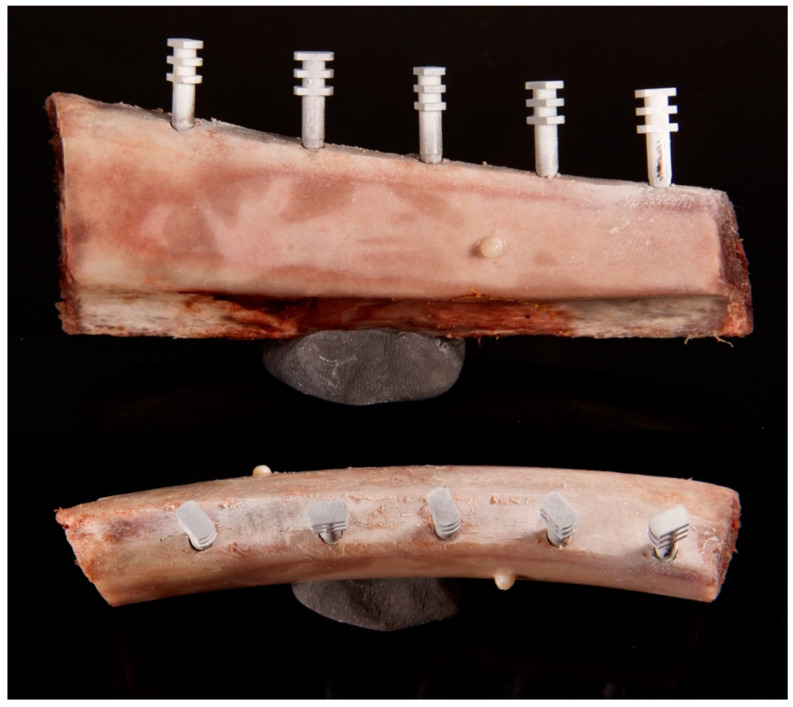
Bone model containing five implants with attached closed tray impression transfers coated with scan spray. Implants were placed with different bone margin thickness. X-ray markers made of dental composite were attached to the bone model surfaces.

**Figure 2 jcm-11-03846-f002:**
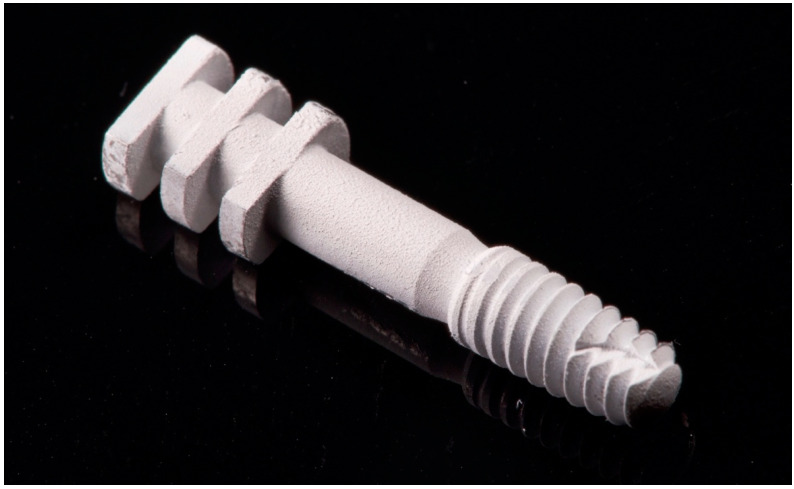
Implant model with an attached closed tray impression transfer prepared for scanning.

**Figure 3 jcm-11-03846-f003:**
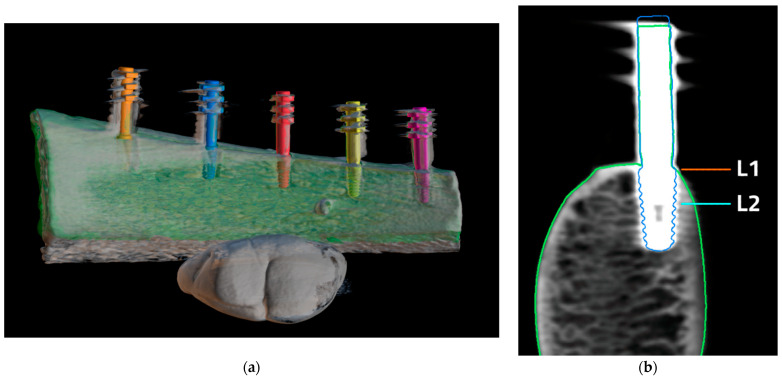
Superimposed STL and DICOM files. (**a**) 3D reconstruction of superimposed DICOM FILES (gray), bone model (green), and implant-with-transfer models (orange, blue, red, yellow, purple); (**b**) Implant cross section, green outline—bone STL model, blue line—implant-with-transfer model. Bone margin measurement levels are marked as L1 (implant neck) and L2 (3.5 mm apically from implant neck).

**Figure 4 jcm-11-03846-f004:**
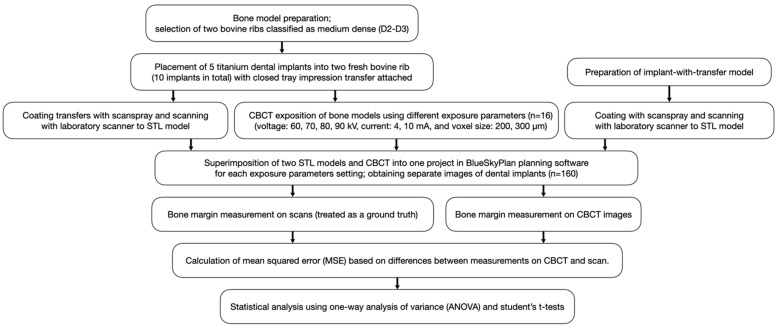
Study design flowchart.

**Table 1 jcm-11-03846-t001:** Measurements on the scans at two bone levels.

Implant	Mean of Two Measurements at Level 1 [mm]	Mean of Two Measurements at Level 2 [mm]
1	0.00	0.17
2	0.27	1.21
3	0.77	1.83
4	1.09	2.31
5	1.20	2.54
6	0.25	1.26
7	0.55	1.55
8	0.61	1.68
9	0.43	1.71
10	0.95	2.19

**Table 2 jcm-11-03846-t002:** Relationship between voltage and MSE of bone thickness measurements.

kV	MSE [mm^2^]	SD of Error [mm^2^]
60	0.27	0.35
70	0.18	0.22
80	0.18	0.19
90	0.14	0.14
All	0.19	0.23

kV—kilovolts, MSE—mean squared error, SD—standard deviation.

**Table 3 jcm-11-03846-t003:** Voxel size and current assessment.

	Current			Voxel Size		
	4 mA	10 mA	*p*-Value	200 µm	300 µm	*p*-Value
mean MSE [mm^2^]	0.18	0.18	0.969	0.20	0.15	0.055

**Table 4 jcm-11-03846-t004:** Linear regression results.

Voltage	Bone Margin Thickness for MSE < 0.25	*p*-Value	Regression Equation
60 kV	never		
70 kV	0.00–0.72	0.161	0.17 × d + 0.13
80 kV	0.00–1.08	0.004	0.21 × d + 0.03
90 kV	0.00–1.12	<0.001	0.23 × d + −0.01
ALL	0.00–0.88	0.047	0.13 × d + 0.13

d—bone margin thickness.

## Data Availability

Data generated or analysed during the study are available from the corresponding author on request.

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
