# Peer review of "Influence of Exposure Parameters and Implant Position in Peri-Implant Bone Assessment in CBCT Images: An In Vitro Study"

_jcm, 2022, doi:10.3390/jcm11133846_

Round 1
Reviewer 1 Report
The present study aimed to assess the impact of dimensional distortion and its changes with modification of exposure setting parameters on the measurements of the bone margin surrounding dental implants. The subject is interesting.
Abstract:
Describe the implant type;
Provide the values of voltage, tube current and voxel size that have been evaluated.
Improve the statistical test description and inform the p-values.
Introduction:
Insert an paragraph explaining the importance of voltage, current, and voxel size and why they should be evaluated as factors in the present study.
What was your study hypothesis?
Methods:
Describe D2-D3 bone types and how the bovine bone was checked to represent them.
What kind of implant connection was used?
“Each implant was placed with different bone margin thickness ranging from 0 mm to 2 mm” please improve.
Figure 1 and 2 should be merged in one figure.
Provide a flowchart showing your study design. It will improve your manuscript didactics.
“Bone margin was measured” How? Describe the software tool used.
“interval between the first and the second reading was at least 6 weeks.” Why?
Discussion:
Discuss the use of Closed tray impression transfer instead scanboody.
“To the best of our knowledge, there are no studies with comparable measuring methods where superimposition of STL models and CBCT images was performed.” Discuss also the disadvantages of performing this approach.
Do you believe similar method can be applied for different abutment materials that present other properties? How they supposed affect the CBCT? Please discuss it. You can check and discuss the reference https://doi.org/10.3390/coatings12020238
Improve your discussion considering the differences between bovine bone model and human bone.
Author Response
Dear Reviewer,
thank you for your comments regarding my article. They help me to improve the quality of the article. I've followed all of your recommendations, edited my article and made the following changes.
Please see the attachment.
Best regards,
Paweł Sawicki, DDS

Reviewer 2 Report
The authors presented an interesting topic and a well-written manuscript
Abstract:
The authors should mentioned some numbers of the results.
Introduction:
The introduction part is well-written and leads the reader to the topic of the manuscript. However, the authors should add the hypothesis which is examined in the presented study.
Material and Methods:
Results:
Discussion:
As the mandible has a “V” or “U” shape, there might be an influence of the beam passing through the “other” side of the mandible. The rib examined in the presented study was a single bone. Have there been examinations considering this topic? Please discuss
Figures:
Good quality and improving the understanding for the reader.
Tables:
Understandable without problems.
Author Response

(The authors gave the same response as above.)
